# DHEA Induces Sex-Associated Differential Patterns in Cytokine and Antibody Levels in Mice Infected with *Plasmodium berghei* ANKA

**DOI:** 10.3390/ijms241612549

**Published:** 2023-08-08

**Authors:** Fidel Orlando Buendía-González, Luis Antonio Cervantes-Candelas, Jesús Aguilar-Castro, Omar Fernández-Rivera, Teresita de Jesús Nolasco-Pérez, Monserrat Sofía López-Padilla, David Roberto Chavira-Ramírez, Armando Cervantes-Sandoval, Martha Legorreta-Herrera

**Affiliations:** 1Laboratorio de Inmunología Molecular, Unidad de Investigación Química Computacional, Síntesis y Farmacología de Moléculas de Interés Biológico, Facultad de Estudios Superiores Zaragoza, Universidad Nacional Autónoma de México, Ciudad de México 09230, Mexico; fidelbuendia2@gmail.com (F.O.B.-G.); cervantescandelasluis@gmail.com (L.A.C.-C.); jesus_aguilar_castro@yahoo.com.mx (J.A.-C.); qfbfdz@gmail.com (O.F.-R.); teresyta.qfb@gmail.com (T.J.N.-P.); sofia.lpz100@gmail.com (M.S.L.-P.); 2Posgrado en Ciencias Biológicas, Unidad de Posgrado, Circuito de Posgrados, Ciudad Universitaria, Universidad Nacional Autónoma de México, Ciudad de México 04510, Mexico; 3Departamento de Biología de la Reproducción, Instituto Nacional de Ciencias Médicas y Nutrición Salvador Zubirán, Ciudad de México 14080, Mexico; roberto.chavira@incmnsz.mx; 4Laboratorio de Aplicaciones Computacionales, Facultad de Estudios Superiores Zaragoza, Universidad Nacional Autónoma de México, Ciudad de México 09230, Mexico; arpacer@unam.mx

**Keywords:** DHEA, *Plasmodium*, gonadectomy, sexual dimorphism, IL-2, IL-4, IFN-γ, IgG

## Abstract

Malaria is the most lethal parasitic disease worldwide; the severity of symptoms and mortality are higher in men than in women, exhibiting an evident sexual dimorphism in the immune response; therefore, the contribution of 17β-estradiol and testosterone to this phenomenon has been studied. Both hormones differentially affect several aspects of innate and adaptive immunity. Dehydroepiandrosterone (DHEA) is the precursor of both hormones and is the sexual steroid in higher concentrations in humans, with immunomodulatory properties in different parasitic diseases; however, the involvement of DHEA in this sexual dimorphism has not been studied. In the case of malaria, the only information is that higher levels of DHEA are associated with reduced *Plasmodium falciparum* parasitemia. Therefore, this work aims to analyze the DHEA contribution to the sexual dimorphism of the immune response in malaria. We assessed the effect of modifying the concentration of DHEA on parasitemia, the number of immune cells in the spleen, cytokines, and antibody levels in plasma of CBA/Ca mice infected with *Plasmodium berghei* ANKA (*P. berghei* ANKA). DHEA differentially affected the immune response in males and females: it decreased IFN-γ, IL-2 and IL-4 concentrations only in females, whereas in gonadectomized males, it increased IgG2a and IgG3 antibodies. The results presented here show that DHEA modulates the immune response against *Plasmodium* differently in each sex, which helps to explain the sexual dimorphism present in malaria.

## 1. Introduction

Malaria is the deadliest parasitic disease; in 2021, it caused 627,000 deaths [1] and is a clear example of sexual dimorphism in which males develop higher mortality than females [2,3]. In general, women exhibit more robust innate and adaptive immune responses against infectious agents and eliminate *Plasmodium* more efficiently than men. Although this sexual condition is accepted, few articles have analyzed this phenomenon in both sexes [4]. 17β-estradiol and testosterone are involved in the sexual dimorphism that occurs in the immune response in malaria [5,6,7]. Both hormones modify survival and cellular and humoral immune responses in *Plasmodium* infections [2,8,9]. DHEA is the precursor hormone of testosterone and 17β-estradiol and has important immunopotentiating properties [10]. DHEA modulates immune response, cell activity and cytokine concentration [11], and this hormone increases macrophage and natural killer (NK) cell activity [12,13] and promotes the maturation of CD4^+^ T cells and their secretion of interferon (IFN) [14]. Furthermore, DHEA boosts the immune response in the bacterial [15], viral and parasitic diseases [16]. In parasitic diseases, DHEA has been shown to promote the immune response in patients infected with *Trypanosoma cruzi*, decreasing parasitemia and TNF-α levels [17]. Furthermore, increased DHEA concentration has been associated with reduced infection by *Schistosoma japonicum* and *Schistosoma mansoni* [18,19]. Additionally, DHEA promotes the Th1-type response and consequently promotes recovery in patients with *Leishmania mexicana* [20]. In the *Plasmodium berghei* infection, the DHEA analog (16α-bromoepiandrosterone) decreases parasitemia in vitro and in vivo [21]. These findings show that DHEA modulates the immune response in parasitic diseases, and all of them exhibit sexual dimorphism [22]. Unfortunately, DHEA participation in the immune response against *Plasmodium*-infected individuals of both sexes has not yet been studied. Sexual dimorphism is a highly complex phenomenon involving sex hormones, genetic differences and even behavior between the sexes [4]. Due to the immunomodulatory properties of DHEA, it is likely that DHEA also contributes to sexual dimorphism in the immune response to *Plasmodium* because together with its sulfated form (DHEA-S), DHEA constitutes the steroid hormone at the highest concentration [23]. Moreover, DHEA positively regulates the immune response [24]. Particularly in malaria, it has been associated with lower parasitemia in *Plasmodium falciparum*-infected individuals and high hemoglobin concentrations [25,26]. However, the mechanisms that explain these findings are unknown. The purpose of this work is to analyze whether DHEA is involved in the sexual dimorphism of the immune response in malaria. To this end, we modified the concentration of this hormone in vivo in male and female mice that were infected with *P. berghei* ANKA; we evaluated the impact on parasitemia, immune cells in the spleen (CD3^+^, CD4^+^, CD8^+^, CD19^+^, macrophages and NK cells), pro- and anti-inflammatory cytokines and IgG and IgM levels. 

To our knowledge, this work is the first report documenting the dimorphic participation of DHEA in the immune response to malaria and contributes to a better understanding of the differences between sexes that could support the future development of sex-specific, more efficient, anti-malarial therapies.

## 2. Results

### 2.1. DHEA Administration Dramatically Increased DHEA Levels in Mice Infected with P. berghei ANKA

We used a previously calibrated dose that increased DHEA concentration in vivo in mice [27]. All DHEA-treated groups showed markedly increased DHEA concentrations regardless of whether they were infected or gonadectomized (Figure 1A,B); furthermore, we detected that infection generated a dimorphic pattern by decreasing DHEA concentrations in intact females compared to males under the same conditions (Figure 1B). 

### 2.2. Effects of Modifying DHEA Concentration on Temperature and Hemoglobin Concentration in Mice Infected with P. berghei ANKA

Quantifying weight loss, hypothermia and decreased hemoglobin levels provides insight into disease severity in CBA/Ca mice infected with *P. berghei* ANKA [6]. We did not detect differences depending on DHEA concentration on body weight; we also did not detect a dimorphic pattern in temperature or hemoglobin concentration (Appendix A). 

### 2.3. DHEA Administration to Intact Male Mice Increased Parasitemia 

Intact females exhibited higher parasitemia than intact males, particularly on days 5 and 6 post infection, inducing a dimorphic pattern (Figure 2A). In contrast to our expectations, DHEA administration to intact mice eliminated the differences between sexes by increasing parasitemia only in intact males (Figure 2B,E). Additionally, gonadectomy did not modify parasitemia in either sex, maintaining the dimorphic pattern (Figure 2C,E). Furthermore, the reconstitution of Gx mice with DHEA increased parasitemia in both sexes on day 6 postinfection (Figure 2D); in addition, the reconstitution of Gx mice with DHEA prior to infection eliminated the dimorphic pattern (Figure 2E). Finally, increasing DHEA administration to 40 mg/kg did not modify parasitemia (Appendix A).

### 2.4. Reducing the DHEA Concentration Generated a Dimorphic Pattern in the CD3^+^, CD19^+^ and NK Cell Populations of Uninfected Mice

Because the spleen is the central organ where the immune system eliminates *Plasmodium*, infection with this parasite induces significant changes in size, architecture and cellular composition [28]. In this work, we evaluated whether modifying DHEA levels affects the central cell populations of the immune response in the spleen. No differences were detected in the CD3^+^ population between uninfected intact males and females; gonadectomy generated a dimorphic pattern in which females had a higher number of CD3^+^ cells than males in the absence of infection. Additionally, the reconstitution of Gx mice with DHEA decreased the number of CD3^+^ cells, which eliminated the dimorphic pattern (Figure 3A). Interestingly, infection decreased this population in all intact groups, and gonadectomy reduced the number of CD3^+^ cells in the infected groups (Figure 3A). Regarding the CD4^+^ cells, gonadectomy reduced this population in uninfected females; however, reconstituting Gx mice with DHEA did not restore their CD4^+^ population, and no changes were detected in infected mice (Figure 3B). Concerning CD8^+^ cells, we detected that intact uninfected females exhibited higher numbers of CD8^+^ cells than males under the same conditions, generating a sex-associated pattern, and DHEA administration further increased this difference. In addition, gonadectomy decreased this population, which eliminated the dimorphic pattern. Moreover, infection completely depleted this population only in the intact groups; in contrast, all Gx and infected groups dramatically increased CD8^+^ numbers regardless of DHEA concentration (Figure 3C).

Regarding the macrophage (CD107b^+^) population, uninfected intact females had a higher number of this population than males under the same conditions, exhibiting a dimorphic pattern. When DHEA was administered to uninfected mice, the dimorphic pattern was maintained; in addition, gonadectomy decreased this cell population only in females, eliminating the dimorphic pattern (Figure 4A). Furthermore, infection increased macrophage numbers in intact vehicle-treated males and DHEA-treated Gx females (Figure 4A).

Regarding B cells (CD19^+^), DHEA administration to intact uninfected mice generated a dimorphic pattern; females exhibited higher numbers of CD19^+^ cells than males under the same conditions; additionally, gonadectomy considerably decreased this population, particularly in females, which also generated a dimorphic pattern that did not change after DHEA reconstitution (Figure 4B). Gonadectomy decreased the number of CD19^+^ cells exclusively in infected females (Figure 4B).

Finally, gonadectomy increased the NK population only in uninfected females, which induced a dimorphic pattern that did not change with DHEA administration. In infected mice, changing the DHEA concentration did not affect this population (Figure 4C).

### 2.5. DHEA Generated a Dimorphic Pattern in the Concentrations of IFN-γ, IL-2 and IL-4

Given that immune response cells produce cytokines and malaria severity is associated with elevated concentrations of TNF-α, IL-6 and IL-10 [29,30]. Additionally, DHEA decreases IFN-γ synthesis and increases IL-10 levels in vitro [31]. Therefore, we assessed the concentrations of pro- and anti-inflammatory cytokines. No differences in IFN-γ concentration were detected between intact male and female mice or Gx mice treated with the vehicle without infection; furthermore, DHEA administration increased IFN-γ concentration only in uninfected females (Figure 5A). In general, infection increased the IFN-γ concentration in intact animals treated with vehicle or DHEA. Administering DHEA decreased the IFN-γ concentration only in intact infected females, which induced a dimorphic pattern (Figure 5A). Regarding TNF-α, in general, males exhibited higher levels than females, although the increase was not significant; the infection increased the TNF-α concentration in the intact groups; however, DHEA administration did not significantly modify the concentration of this cytokine (Figure 5B). Concerning IL-2, gonadectomy in uninfected mice resulted in a dimorphic pattern, with males having higher IL-2 concentrations than females under the same conditions. Reconstituting Gx mice with DHEA dramatically decreased the IL-2 concentration in uninfected Gx males, eliminating the dimorphic pattern. In infected mice, gonadectomy increased the IL-2 concentration only in vehicle-treated females, and reconstitution of Gx mice with DHEA significantly decreased the IL-2 concentration exclusively in females, resulting in a dimorphic pattern (Figure 5C).

Regarding IL-10, modifying the concentration of DHEA did not change the concentration of this cytokine in uninfected mice. In addition, infection increased the concentration of IL-10 in intact females, generating a dimorphic pattern in which females exhibited a higher concentration of IL-10 than males. DHEA administration eliminated this dimorphic pattern. Furthermore, gonadectomy decreased the concentration of IL-10 only in infected females, and reconstitution of Gx females with DHEA did not restore the concentration of this cytokine in infected females (Figure 5D).

Related to IL-4, DHEA-reconstituted Gx mice in the absence of infection had decreased IL-4 concentrations, whereas, in infected mice, DHEA-reconstituted male Gx mice had higher IL-4 concentrations than females under the same conditions, resulting in a sex-dependent pattern (Figure 6A). Concerning IL-6, the reconstitution of Gx mice with DHEA without infection increased IL-6 concentrations in both sexes. However, neither gonadectomy nor infection changed the IL-6 concentration (Figure 6B). Finally, infection or DHEA administration did not change the IL-17 concentration (Figure 6C).

### 2.6. Reconstitution of Gx Males with DHEA Increased the Concentration of the IgG Subclass

Antibodies are required to eliminate the *Plasmodium* parasite [32]; in addition, DHEA has been reported to regulate antibody synthesis in mice [33]; thus, we tested whether modifying DHEA concentration affects antibody levels. DHEA administration did not significantly modify the IgM concentration or total IgG in the intact or Gx mice (Figure 7A,B). Nevertheless, when we evaluated IgG subclasses, we detected that the reconstitution of gonadectomized mice with DHEA increased the levels of IgG1, IgG2a, IgG2b and IgG3 exclusively in males (Figure 7C–F). To determine whether the immune response was polarized to a Th1 or Th2 type, the IgG1/IgG2a ratio was assessed. As the ratio in all groups was lower than one, DHEA promoted a Th1 response (Figure 7G).

## 3. Discussion 

To evaluate the participation of DHEA in the sexual dimorphism that occurs in the immune response against *P. berghei* ANKA, we modified the concentration of DHEA in male and female mice. We decreased its levels by gonadectomy or increased it by exogenous administration. After treatments, mice were infected with *P. berghei* ANKA. We found that gonadectomy, infection and DHEA administration generated sex-associated patterns.

We first demonstrated that the dose of DHEA we administered increased its concentration in plasma. This effect is consistent with that described by Medina et al., who reported that subcutaneous DHEA administration increases the concentration of this steroid up to 24 times on day 7 post treatment [34]. However, in our work, gonadectomy only decreased DHEA concentration by a trend; a likely explanation for this finding is that since gonadectomy was performed four weeks prior to infection, the mouse censored its deficiency and was able to synthesize DHEA at extragonadal sites [35]. In addition, we found that DHEA did not change body weight, temperature, or hemoglobin concentration. However, gonadectomy decreased the temperature in the uninfected groups in both sexes. It is possible that this surgery, by decreasing the concentration of 17β-estradiol [7], deregulated thermogenesis in adipose tissue [36]. In addition, DHEA increases mitochondrial FADH_2_ synthesis, resulting in heat generation [37]. In a further finding, gonadectomy increased hemoglobin concentration, which is an unexpected result because high levels of DHEA are associated with increased hemoglobin concentration [26]. This increase is likely to be a consequence of the reduction in 17β-estradiol concentration, as it negatively regulates the erythropoietin synthesis [38]. In addition, gonadectomy in male rats has been reported to increase erythropoietin levels [39]. We also analyzed the effect of modifying DHEA concentration on parasitemia, detecting that intact and Gx-infected females treated with vehicle had higher parasitemia than males; interestingly, this dimorphic pattern was eliminated when DHEA was administered, as it increased parasitemia in intact males without affecting female mice; then, we tried a higher dose (40 mg/kg), but the result was the same (Appendix A). To our knowledge, ours is the first report that DHEA increases parasite load. In contrast, Freilich et al. reported that the DHEA analog 16α-bromoepiandrosterone decreases parasitemia in *P. berghei* ANKA-infected rats [21], and it is likely that this difference is due to DHEA presenting higher affinity for the androgen receptor than 16α-bromoepiandrosterone [40]. Our result also contrasts with that described by dos Santos et al. in *Trypanosoma cruzi* infection [41]. A probable explanation is the parasite used, as we experimented with *P. berghei* ANKA in an in vivo study. In addition, it is likely that these discrepancies are because immune response cells have different numbers of receptors for DHEA in males than in females [42], which also vary depending on whether the mice are intact or gonadectomized [43].

Because the spleen is the leading site of *Plasmodium* destruction, we analyzed the effect of modifying DHEA levels on the central cells of the immune response in the spleen. We found that uninfected Gx females exhibited higher numbers of CD3^+^ cells than males under the same conditions; this result corresponds with the dimorphic pattern found in IL-2 concentration in this group. Most likely, IL-2, in addition to promoting CD3^+^ cell proliferation, it also regulates the process of apoptosis mediated by FASL expression in T cells [44]. This explanation is supported by the disappearance of the dimorphic pattern in T cells when DHEA was administered to Gx mice. In contrast, Cao et al. described that administering DHEA increases both T-lymphocyte proliferation and IL-2 concentration [13]. This difference could be explained because the Cao group administered a dose of 14.42 µg/kg DHEA orally, while we used 8 mg/kg DHEA subcutaneously. Furthermore, infection decreased the number of CD3^+^ cells in intact vehicle- or DHEA-treated females and vehicle-treated Gx females compared to their uninfected controls. A likely explanation is that *Plasmodium* infection induces apoptosis in this cell population [45].

In this study, uninfected intact females had higher numbers of CD3^+^/CD8^+^ cells than males under the same conditions, which corroborates the findings of Arsenovic et al., who demonstrated that female rats have higher numbers of CD8^+^ T lymphocytes than males [46]. We show that DHEA administration increased the number of CD3^+^/CD8^+^ cells in uninfected female mice, further accentuating the dimorphic pattern of this population. A probable explanation is that DHEA at high concentrations binds to ERα and Erβ (estrogen receptors) [47], and this interaction induces the proliferation of T-lymphocytes [48]. However, this possibility requires experimental demonstration in our model. Moreover, infection decreased the number of CD3^+^/CD8^+^ cells in intact males and females, corresponding with the decrease in total CD3^+^ cells during infection. This result is likely due to *Plasmodium* infection inducing apoptosis of CD3^+^/CD8^+^ T cells in the spleen [49]. Another possibility is that TCD3^+^/CD8^+^ lymphocytes migrate to lymphoid nodules or to the brain [50,51,52]. In addition, modifying the DHEA concentration did not change the number of macrophages in any group, which is probably why DHEA did not decrease parasitemia in our malaria model. Regarding CD19^+^ cells, DHEA administration did not modify this population; nevertheless, gonadectomy decreased it in both uninfected and infected females treated with vehicle. This effect is probably due to the loss of estrogen regulation of CD19^+^ cells by removing the gonads, as previously shown [6,53].

Finally, gonadectomy induced a dimorphic pattern in the NK cell population, with Gx-uninfected male mice showing higher numbers of NK cells than females under the same conditions. This dimorphic pattern may be associated with Gx males developing higher IL-2 concentrations than Gx females, as IL-2 is one of the main factors promoting NK cell proliferation [54]. In summary, DHEA administration decreased the concentration of IL-2 in Gx females, which increased the number of CD3^+^/CD8^+^ T cells.

Immune cells modulate immunity by secreting cytokines. In this work, we detected that infection with *P. berghei* ANKA increased the concentrations of IFN-γ and TNF-α, which corroborates what has been previously described [55]. In addition, DHEA administration decreased the concentration of IFN-γ in intact infected females compared to intact infected males, resulting in a sex-associated pattern. This pattern corresponds to the lower concentration of IFN-γ in DHEA-treated female rats infected with *Trypanosoma cruzy* compared to male rats under the same conditions [56], which indicates that DHEA modulates cytokine production differently in males and females. Furthermore, DHEA inhibits the inflammatory response by preventing NF-κB translocation and PI3K-mediated signaling [57], which promotes the activation of lymphocytes. Finally, it is also likely that DHEA decreases IFN-γ secretion by modulating NK cells, which are the main IFN-γ expressing cells in the spleen [49].

On the other hand, gonadectomy increased the IL-2 concentration in males compared to females without infection, resulting in a dimorphic pattern. Interestingly, this sex-associated pattern was eliminated when DHEA was administered, which corroborates the findings of Pratschke et al. who reported that DHEA administration decreases IL-2 synthesis in lymphocytes from patients with abdominal surgery. [58]. In contrast, in the infected groups, the reconstitution of Gx mice with DHEA decreased the IL-2 concentration in females only, resulting in a sex-dependent pattern.

Furthermore, infection increased the IL-10 concentration exclusively in intact females, resulting in a dimorphic pattern that was eliminated by DHEA administration. Interestingly, this group had the lowest DHEA concentration and the highest IL-10 levels. This finding corresponds with that described in patients with systemic lupus erythematosus in whom DHEA administration decreases their IL-10 concentration [59]. Another possible explanation is the reciprocal regulation between the concentration of the cytokines IFN-γ and IL-10 in *Plasmodium* infection [60], which is relevant given that IL-10 decreases the incidence of cerebral malaria and modulates the effects of IFN-γ [61,62]. Moreover, Omer and Riley demonstrated that TGF-β, in addition to increasing IL-10 concentration, improves survival and decreases parasitemia in *Plasmodium berghei* infected mice [63]. Given that Treg cells are important for regulating inflammation by producing TGF-β [64] and that DHEA modulates the proliferation of Tregs [65], it would be important to evaluate the effect of DHEA on TGF-β concentration as well as to assess the population of Tregs to understand the immunoregulation exerted by DHEA in this disease.

An interesting finding is that modifying the concentration of DHEA in infected mice only affected the concentration of IL-2 and IL-4 in Gx females, which generated dimorphic patterns depending on sex, a likely explanation being that IL-2 is required for IL-4 synthesis [66]. In addition, DHEA administration decreased the IL-4 concentration in uninfected Gx males, which was expected given that this steroid suppresses the Th2-type response [13,67]. Regarding IL-6, DHEA administration increased the levels of this cytokine in uninfected Gx male and female mice, an effect that differs from that reported by Sudo et al., who described that DHEA administration suppressed IL-6 levels [33]. The discrepancy in this result is probably because we performed the determination in Gx mice, and hormones produced in the gonads, such as testosterone, also upregulate the synthesis of this cytokine [68].

Related to the antibody levels, reconstituting Gx and infected males with DHEA increased the concentration of IgG1, IgG2a, IgG2b and IgG3 antibodies in plasma. This finding corroborates the results of Cheng and Tseng, who described that DHEA administration increases the concentrations of IgM and IgG [69]. In addition, regardless of treatment or sex, the IgG1/IgG2a ratio was less than one, suggesting a predominance of the Th1-type response characteristic of *P. berghei* infections [70]. However, only DHEA-treated and infected Gx males exhibited increased IgG1, IgG2a, IgG2b and IgG3 levels. A probable explanation is that DHEA decreased the IL-4 concentration in Gx females under the same conditions, and this cytokine is a growth and differentiation factor for B lymphocytes that promotes their differentiation into antibody-producing cells [71].

The increase in antibody concentration induced by DHEA is important because it is associated with reduced disease severity [72]. It is critical to note that gonadectomy and DHEA administration could also affect testosterone and 17β-estradiol levels, which could influence the interpretation of the results.

While the molecular mechanisms by which DHEA modifies immune function in malaria remain to be revealed, Vargas-Villavicencio et al. have shown that DHEA administration increases androgen receptor (AR) expression in the spleen [27]. In addition, DHEA also binds estrogen receptors ERα and ERβ [40], suggesting that DHEA interacts with the AR, ERα and ERβ present in immune response cells and acts through mechanisms involving a classical nuclear receptor in the immune system, although the affinity for these receptors is low [73]. In addition, DHEA may also be transformed into androgens, estrogens or other metabolites in peripheral tissues [74]. 

## 4. Materials and Methods

### 4.1. Mice 

Four-week-old male or female CBA/Ca mice were used. The mice were a generous gift from Dr. William Jarra (National Institute for Medical Research, London, UK). The animals were bred and maintained in a specific pathogen-free environment in the animal house of the Facultad de Estudios Superiores Zaragoza, Universidad Nacional Autónoma de México. All protocols used in animal handling were authorized by the animal care and use committee, registration number 24/08/SO/3.4.1, respecting the official national standard NOM-062-ZOO-1999.

### 4.2. Gonadectomy

#### 4.2.1. Orchidectomy 

Male mice were gonadectomized (Gx) as previously described [8]. In brief, 4-week-old CBA/Ca male mice were anesthetized with a mixture of ketamine: xylazine 80 mg/kg: 8 mg/kg (Phoenix Pharmaceutical Inc., St. Joseph, MO, USA). The testes and epididymis were removed via scrotal excision under sterile conditions, and the efferent ducts were sealed by electrocautery.

#### 4.2.2. Ovariectomy

Four-week-old female CBA/Ca mice were anesthetized with the mixture (ketamine:xylazine), incisions were made in the lower abdomens under aseptic conditions, ovaries were removed, and muscles were sutured. Male and female mice were used 4 weeks after surgery to allow for their recovery.

### 4.3. DHEA Administration

DHEA (Sigma-Aldrich, St. Louis, MO, USA) was diluted in vehicle (sweet almond oil LASA, México City, Mexico) and injected subcutaneously (8 mg/kg body weight) daily for 5 days prior to infection as previously described [27]. Control groups received 50 μL of vehicle.

### 4.4. Parasite, Infection and Parasitemia

*P. berghei* ANKA was kindly donated by Dr. William Jarra (National Institute for Medical Research, London, UK). Parasites were expanded in mice and stored in cryovials under liquid nitrogen. To activate the parasite, a vial was thawed at room temperature and immediately intraperitoneally inoculated into a 4-week-old mouse. When parasitemia reached approximately 20%, a blood sample was drawn in cold PBS, the total number of erythrocytes was counted in a Neubauer chamber and blood was diluted with PBS to obtain a suspension with 1 × 10^4^ parasitized red cells/mL. For infection, each mouse was inoculated intravenously with 100 µL of the above suspension (1 × 10^3^ parasitized red cells).

To evaluate parasitemia, blood smears were prepared, fixed with absolute methanol and stained with Giemsa (Sigma-Aldrich, St. Louis, MO, USA) diluted 1:10 in phosphate buffer. To quantify the number of parasites, the 100× objective of a Carl Zeiss Standard 20 Microscope (Carl Zeiss Ltd., Welwyn Garden City, UK) was used. Fifty fields were evaluated when parasitemia was less than 2%, and 200 erythrocytes were evaluated when parasitemia exceeded this number. Since *P. berghei* ANKA is lethal and most mice die around day 9 post infection, we sacrificed all mice on day 8 post infection [8]. Parasitemia in each group was presented as the geometric mean of parasitized red cells ± SEM.

### 4.5. Body Weight and Temperature

Daily from day 0 to day 8 postinfection, all mice were weighed on a semianalytical balance (Ohaus, Parsippany, NJ, USA). Body temperature was also assessed daily using an infrared thermometer (Thermofocus, 01500A/H1N1, Vedano Olona-Varse, Italy).

### 4.6. Hemoglobin Concentration

Two microliters of mouse tail blood were collected daily and mixed with 498 µL of Drabkin solution (1 mM potassium ferrocyanide, 7.6 mM potassium cyanide and 11.9 mM sodium bicarbonate (Sigma-Aldrich)). The mixture was incubated at room temperature for 5 min. The absorbance was measured at 540 nm on a plate reader (Multiskan GO, Thermo Fisher Scientific, Inc., Waltham, MA, USA). To calculate the hemoglobin concentration, a curve was prepared with a rat hemoglobin standard (Sigma-Aldrich).

### 4.7. Extraction of Sex Hormones from Plasma Samples

On day 8 post infection, mice were sacrificed, and blood was collected into heparinized tubes and centrifuged at 2000× *g* for 15 min. Plasma was stored in aliquots at −70 °C until use. For extraction of sex steroids, 100 μL of plasma, 900 µL of PBS and 5 mL of ethyl ether (JT Baker, Fisher Scientific SL, Phillipsburg, NJ, USA) were mixed vigorously for 5 min. The aqueous phase was frozen in a dry ice bath with 96% ethanol. The organic phase was transferred to a glass tube and evaporated in a water bath at 37 °C for 48 h. Samples were diluted with 1000 µL of 0.1% PBS/gelatin (Sigma-Aldrich).

### 4.8. Quantification of DHEA in Plasma

The commercial DHEA kit EIA-3415 (DRG International, Springfield NJ, USA) was used. Ten microliters of plasma extract were mixed with 200 µL of DHEA-conjugated horseradish peroxidase and incubated for one hour at room temperature. The plates were washed 4 times. and 100 µL of tetramethylbenzidine (H_2_O_2_-TMB 0.26 g/L) was added. The plates were incubated for 15 min at room temperature protected from light. The reaction was halted with stop solution, and the plates were gently shaken and read at 450 nm on a Multiskan Ascent 96 plate reader (Thermo Fisher Scientific, Waltham, MA, USA). The DHEA concentration was calculated using a standard curve included in the kit.

### 4.9. Quantification of Cell Populations in the Spleen

Quantification of cell populations was performed by flow cytometry as previously described [7]. Briefly, on day 8 post infection, mice were sacrificed, the spleen was disaggregated on nylon mesh, cells were washed with PBS and erythrocytes were removed with a lysis solution (Beckton and Dickinson, Franklin Lakes, NJ, USA). Cells were washed with sterile staining buffer (PBS, 1% bovine albumin, 0.1% NaN_2_), and the cells were stained with the following dilutions of fluorochrome-coupled anti-mouse antibodies: CD3-FITC clone 17A2 (1:250), CD4-APC clone RM4-5 (1:1000), CD8-PE clone 53-6.7 (1:1000), CD107b-PE clone M3/84 (1:125), CD19-APC clone 1D3/CD19 (1:1600) and CD16/32-PE clone 93 (1:150). We used three antibody combinations: the first for total T cells (CD3^+^), T-helper cells (CD3^+^/CD4^+^) and cytotoxic T cells (CD3^+^/CD8^+^). The second combination identified B cells and macrophages (CD19^+^ and CD107b), respectively, and the third (CD19^−^CD3^−^CD16^+^/CD32^+^) identified NK cells. The cells were incubated for 30 min at room temperature and protected from light. The cells were then washed with PBS and suspended in 100 µL of FACS solution and acquired on a FACS Aria II cytometer (BD Biosciences, San Jose, CA, USA). From this region, we selected the CD3^+^ population by plotting SSC vs FITC-CD3^+^ and calculated the percentage positive for the first gate (to exclude CD19^+^ and NK cells). Two dot plots corresponding to CD4^+^ and CD8^+^ cells were generated from the first region (SSC vs. APC-CD4^+^ and SSC vs. PE-CD8^+^), and the percentages were calculated according to the lymphocytes in the first gate. The sum of CD4^+^ and CD8^+^ lymphocyte percentages is the total CD3^+^ lymphocyte count. In the second combination, CD19^+^ and CD107b cells were identified with APC-antiCD19 and PE-antiCD107b, respectively. The CD19^+^ cells were then selected using the SSC dot plot vs APC-CD19. Macrophages were selected based on the FSC and SSC dot blot, which corresponded to 100%. This procedure excludes CD3^+^, CD19^+^ and NK cells from macrophages that were selected by plotting SSC vs. PE-antiCD107b. Finally, the third combination was used to identify NK cells and four dot plots were used: in the first, we selected lymphocytes and NK cells by drawing SSC vs. FSC; in the second dot blot we used FITC-antiCD3 to select the CD3^−^ population; in the third dot blot, we plotted APC-antiCD19 and selected CD19^−^ cells; and in the fourth dot blot, we plotted SSC vs PE-CD16^+^/32^+^. The first step to analyze the results was to define the negative and positive regions of the fluorescence in each dot plot, using the control of cells stained with a single antibody. Subsequently, the fluorescence of each tube containing only one fluorochrome and the isotype controls were used to determine the positive region for each antibody. For each mixture, 10,000 cells were acquired. All antibodies were purchased from BioLegend (San Diego, CA, USA). Data were analyzed using FlowJo 2.5.1 software (Beckton and Dickinson, Ashland, OR, USA).

### 4.10. Plasma Th1/Th2/Th17 Cytokine Quantification

Groups of intact males or females or Gx mice were treated with DHEA and infected with *P. berghei* ANKA. On day 8 post infection, blood was drawn and plasma was separated and frozen at −70 °C until use. The cytokine levels of IFN-γ, TNF-α, IL-2, IL-4, IL6, IL-10 and IL-17 were quantified in plasma using the Cytometric Bead Array (CBA) Mouse (BD Mouse Th1/Th2/Th17 cytokine CBA Kit Biosciences-Pharmingen, Heidelberg, Germany). Briefly, 25 µL of plasma or 25 µL of the standards were incubated with 50 μL of beads coated with antibodies directed against IFN-γ, TNF-α, IL-2, IL-4, IL-6, IL-10 and IL-17. Each sample was incubated with 50 μL of detection reagent (phycoerythrin) for 2 h at room temperature and protected from light. One milliliter of wash solution was added, and samples were centrifuged at 200× *g* for 5 min. The supernatant was removed, and the beads were suspended in 100 μL of wash solution. Finally, the samples were evaluated in a FACSAria II cytometer, and the results were analyzed with the FCAP array program. A standard curve was used to quantify the concentration of each cytokine. The detection levels in the kit used were as follows: IL-2 (0.1 pg/mL), IL-4 (0.03 pg/mL), IL-6 (1.4 pg/mL), IL-17 (0.8 pg/mL), IFN-γ (0.5 pg/mL) and TNF (0.9 pg/mL).

### 4.11. Quantification of IgM and IgG Antibody Concentrations

Antibody levels were assessed as described previously [75]. Briefly, 96-well plates (Corning, NY, USA) were incubated with 100 µL of *P. berghei* ANKA antigen solution (10 μg/mL) and blocked with 10% serum at 4 °C overnight. The plate was washed and incubated with the test plasma samples for 2 h at 37 °C. Pre calibrated dilutions of biotinylated antibodies specific for IgM, IgG, IgG1, IgG2a, IgG2b and IgG3 (Zymed, South San Francisco, CA, USA) diluted in PBS tween 0.05% milk 0.02% were added and incubated for 1 h at 37 °C. The plate was washed and incubated with streptavidin–peroxidase solution (Sigma-Aldrich) diluted 1:2500 in PBS tween 0.05% for 1 h at 37 °C. After washing, the plate was incubated with 100 μL orthophenylenediamine (OPD) and 0.03% H_2_O_2_ diluted in citrate buffer (Sigma-Aldrich). Finally, the reaction was halted with 100 µL of 0.1 N H_2_SO_4_, and the absorbance at 450 nm was measured on a Multiskan GO plate spectrophotometer (Thermo-Fisher Scientific, Inc.). Since there are no commercial standards to be used as positive controls, optical density at 450 nm was used to denote antibody levels.

### 4.12. Statistical Analysis

Parasitemia, body weight, temperature, hemoglobin concentration, steroid concentration, cell populations, cytokine concentration and antibodies, for calculations two-way ANOVA was performed using Statgraphics XVI (The Plaines, VA, USA) *p* ≤ 0.05 with a Bonferroni post hoc test *p* ≤ 0.05. 

## 5. Conclusions 

In contrast to our expectations, DHEA increased parasitemia in intact males, and its immunomodulatory effects depended on sex, parasite infection and DHEA concentration. Our results explain, at least in part, the sexual dimorphism in the immune response in malaria since DHEA administration decreased IFN-γ and IL-10 concentrations only in intact females. In contrast, in gonadectomized males, DHEA increased the number of CD8^+^ T cells without infection. Interestingly, in infected Gx males, DHEA increased IgG1, IgG2a, IgG2b, IgG3 and IL-4 levels, corresponding to both Th1 and Th2 responses (Figure 8). However, the molecular mechanisms involved are unknown. Therefore, it is essential to explore in depth the protective mechanisms of DHEA in cerebral malaria, where a Th2-type response could counteract the inflammatory process of this complication. Finally, this work helps to explain at least in part the sexual dimorphism of the immune response in malaria, which will be useful for the future development of therapies that are specific to men and women, as they respond differently to DHEA administration.

## Figures and Tables

**Figure 1 ijms-24-12549-f001:**
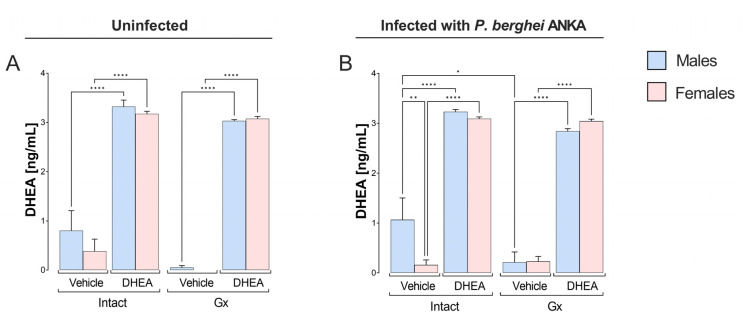
Effect of modifying DHEA levels on plasma steroid concentrations in male and female mice infected with *P. berghei* ANKA. Eight groups of male mice and eight groups of female mice were used; four groups of each sex were gonadectomized (Gx), and the other four groups remained intact as controls. Four weeks after surgery, two Gx and 2 intact groups were treated with DHEA for 5 days. The day after the last administration, one group of Gx and one of the intact mice of each sex were infected with *P. berghei* ANKA, and the remaining groups were injected with PBS as infection control groups. On day 8 post infection or on the equivalent day in uninfected groups, the mice were sacrificed, heart blood was extracted, and plasma was separated to determine the plasma concentration of DHEA. Uninfected groups are represented in (**A**) and infected groups in (**B**). The histogram represents the mean of each group ± SEM mean standard error (*n* = 10). Lines above the histogram represent significant differences between groups. Asterisks (*) indicate statistical significance between 2 groups * (*p* ≤ 0.05), ** (*p* ≤ 0.01), and **** (*p* ≤ 0.0001). The significance between groups was calculated with two-way ANOVA with a Bonferroni post hoc test. The whole experiment was performed twice.

**Figure 2 ijms-24-12549-f002:**
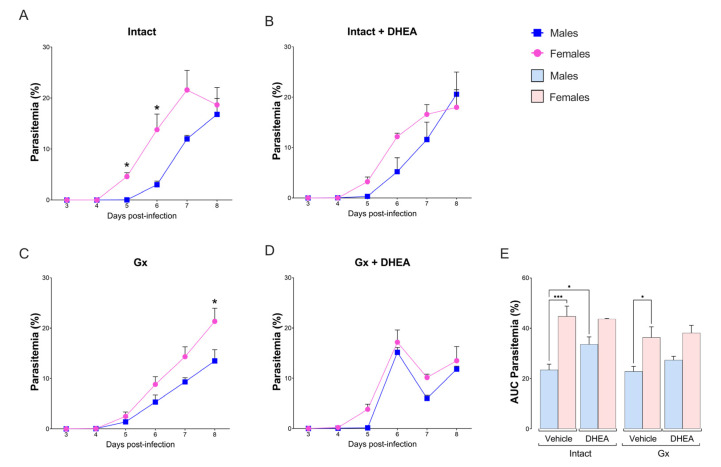
Effect of DHEA on parasitemia in male and female mice infected with *P. berghei* ANKA. Four groups of male mice and four groups of female mice were used; two groups of each sex were gonadectomized (Gx), and the other two groups remained intact as controls. Four weeks after surgery, two Gx and 2 intact groups were treated with DHEA for 5 days. The day after the last administration, one group of Gx and one intact group of each sex were infected with *P. berghei* ANKA (day zero), and the remaining groups were injected with PBS as infection control groups. Parasitemia was assessed daily from day 3 post infection. Parasitemia of intact male and female mice are shown in (**A**); parasitemia of DHEA-treated mice in (**B**); parasitemia of Gx mice in (**C**); parasitemia of Gx DHEA-treated mice in (**D**); and the area under the curve (AUC) of the above groups is shown in (**E**). Graphs represent the mean of each group ± SEM (*n* = 10). Asterisk (*) indicates statistical significance between 2 groups (*p* ≤ 0.05) and *** (*p* ≤ 0.001) calculated with two-way ANOVA and Bonferroni post hoc test. The whole experiment was performed twice.

**Figure 3 ijms-24-12549-f003:**
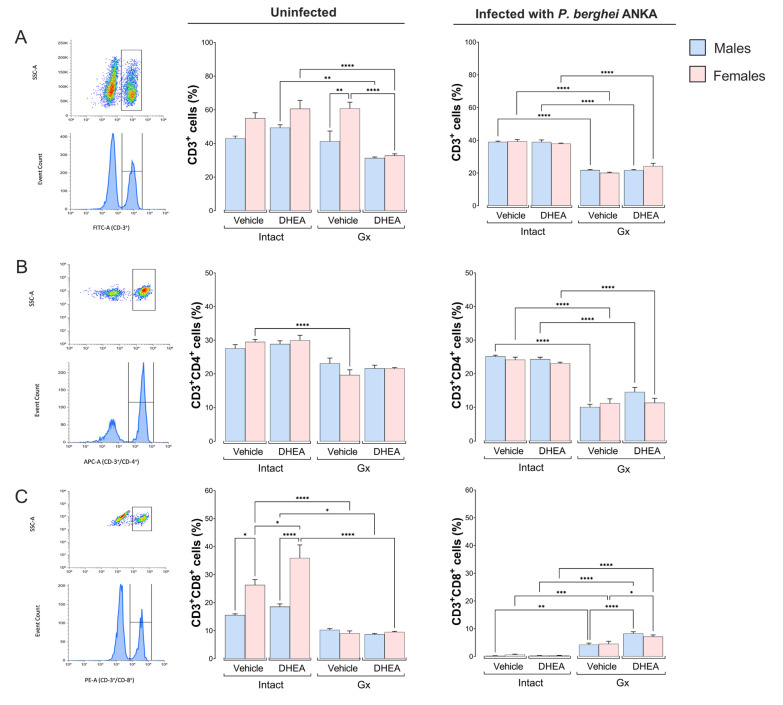
Effect of modifying DHEA levels on CD3^+^, CD3^+^/CD4^+^ and CD3^+^/CD8^+^ cell populations of male and female mice, uninfected and infected with *P. berghei* ANKA. Eight groups of male mice and eight groups of female mice were used; four groups of each sex were gonadectomized (Gx), and the other four groups remained intact. Four weeks after surgery, two Gx and 2 intact groups were administered DHEA for 5 days. The day after the last administration, one group of Gx and one group of intact mice of each sex were infected with *P. berghei* ANKA, and the remaining groups were injected with PBS as infection control groups. On day 8 postinfection, mice were sacrificed, spleens were removed and immune response cells were evaluated by flow cytometry. In the dot plot, the rectangle frames 100% of the cells positive for CD3^+^, CD4^+^ or CD8^+^ markers. The bar graphs represent the average number of cells in each group ± SEM. CD3^+^ lymphocytes are represented in (**A**), helper T cells (CD4^+^) in (**B**), cytotoxic T cells (CD8^+^) in (**C**). Lines above histograms represent significant differences between groups (*n* = 10). Asterisks (*) indicate statistical significance between groups * (*p* ≤ 0.05), ** (*p* ≤ 0.01), *** (*p* ≤ 0.001) and **** (*p* ≤ 0.0001). The significance between groups was calculated with two-way ANOVA with a Bonferroni post hoc test. The whole experiment was performed twice.

**Figure 4 ijms-24-12549-f004:**
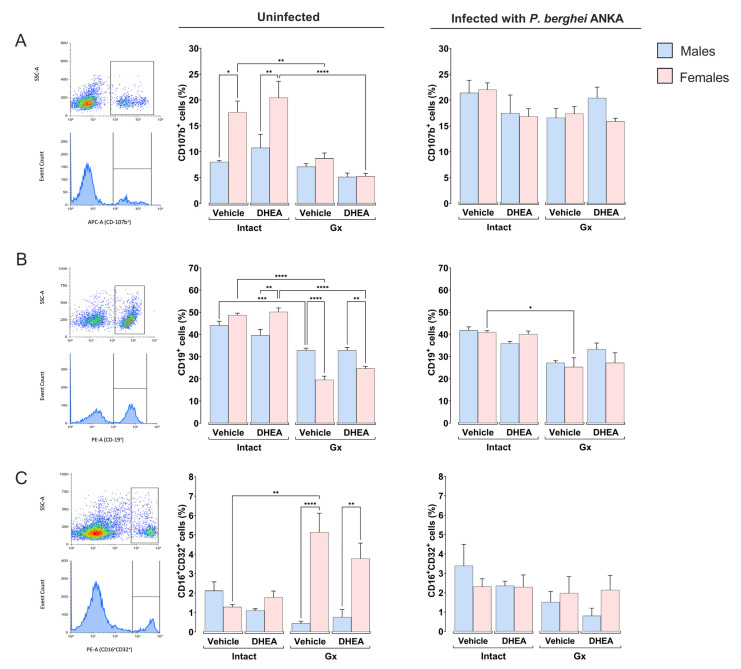
Effect of modifying DHEA levels on macrophage (CD107b^+^), B-cell (CD19^+^) and NK cell (CD3^−^CD16^+^/CD32^+^) populations of male and female mice, uninfected and infected with *P. berghei* ANKA. Eight groups of male mice and eight groups of female mice were used; four groups of each sex were gonadectomized (Gx), and the other four groups remained intact. Four weeks after surgery, two Gx and 2 intact groups were administered DHEA for 5 days. The day after the last administration, one group of Gx and one group of intact mice of each sex were infected with *P. berghei* ANKA, and the remaining groups were injected with PBS as infection control groups. On day 8 post infection, mice were sacrificed, spleens were removed and immune response cells were evaluated by flow cytometry. In the dot plot, the rectangle frames 100% of the cells positive for CD107b^+^ or CD19^+^ or CD16^+^/CD32^+^ markers. The bar graphs represent the average number of cells in each group ± SEM. Macrophages (CD107b^+^) are represented in (**A**), B cells (CD19^+^) in (**B**), and NK cells (CD3^−^/CD19^−^/CD16^+^/CD32^+^) in (**C**). Lines above histograms represent significant differences between groups (*n* = 10). Asterisks (*) indicate statistical significance between 2 groups * (*p* ≤ 0.05), ** (*p* ≤ 0.01), *** (*p* ≤ 0.001) and **** (*p* ≤ 0.0001). The significance between groups was calculated with two-way ANOVA with a Bonferroni post hoc test. The whole experiment was performed twice.

**Figure 5 ijms-24-12549-f005:**
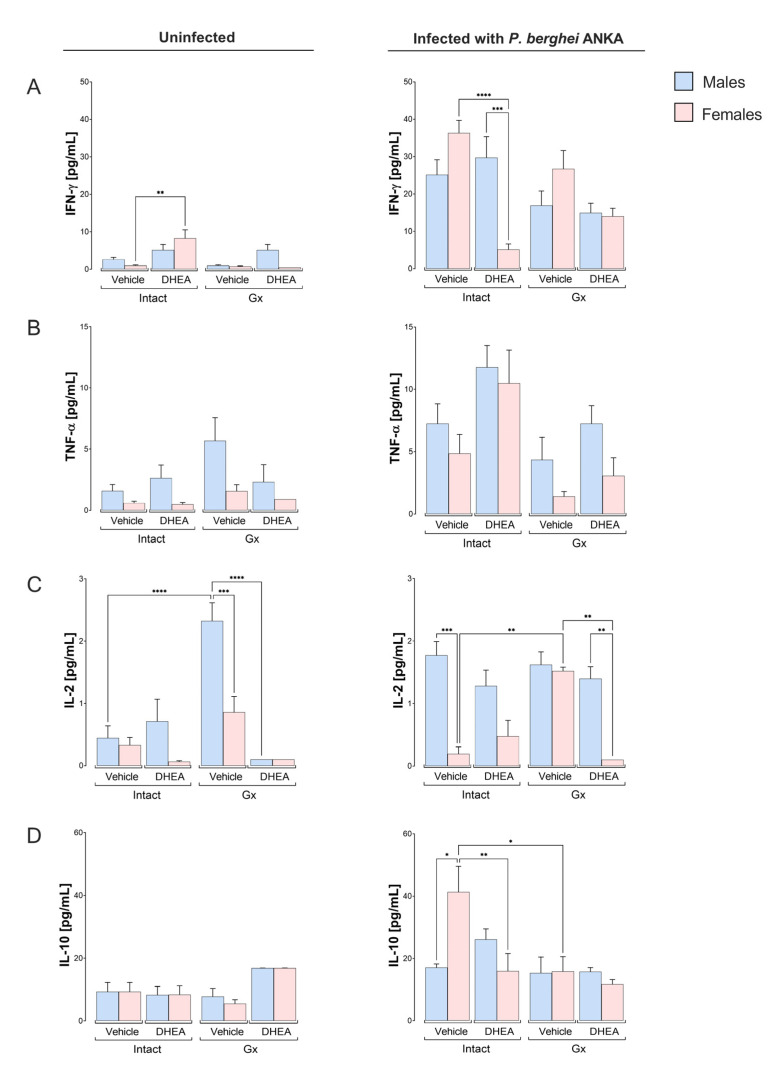
Effect of modifying DHEA levels on the plasma concentration of cytokines: IFN-γ, TNF-α, IL-2 and IL-10 in uninfected and infected male and female mice with *P. berghei* ANKA. Eight groups of male mice and eight groups of female mice were used; four groups of each sex were gonadectomized (Gx), and the other four groups remained intact. Four weeks after surgery, two Gx and two intact groups were reconstituted with DHEA for 5 days. On the day following the last administration, one group of Gx and one of the intact mice of each sex were infected with *P. berghei* ANKA, and the remaining groups were injected with PBS as infection control groups. On day 8 post infection, mice were sacrificed and plasma was obtained to quantify the levels of the cytokines IFN-γ (**A**), TNF-α (**B**), IL-2 (**C**) and IL-10 (**D**). Determination was performed by flow cytometry. Histograms represent the mean of each group ± SEM (*n* = 10). The lines above histograms represent significant differences between groups. Asterisks (*) indicate statistical significance between 2 groups * (*p* ≤ 0.05), ** (*p* ≤ 0.01), *** (*p* ≤ 0.001) and **** (*p* ≤ 0.0001). The significance between groups was calculated with two-way ANOVA with a Bonferroni post hoc test. The whole experiment was performed twice.

**Figure 6 ijms-24-12549-f006:**
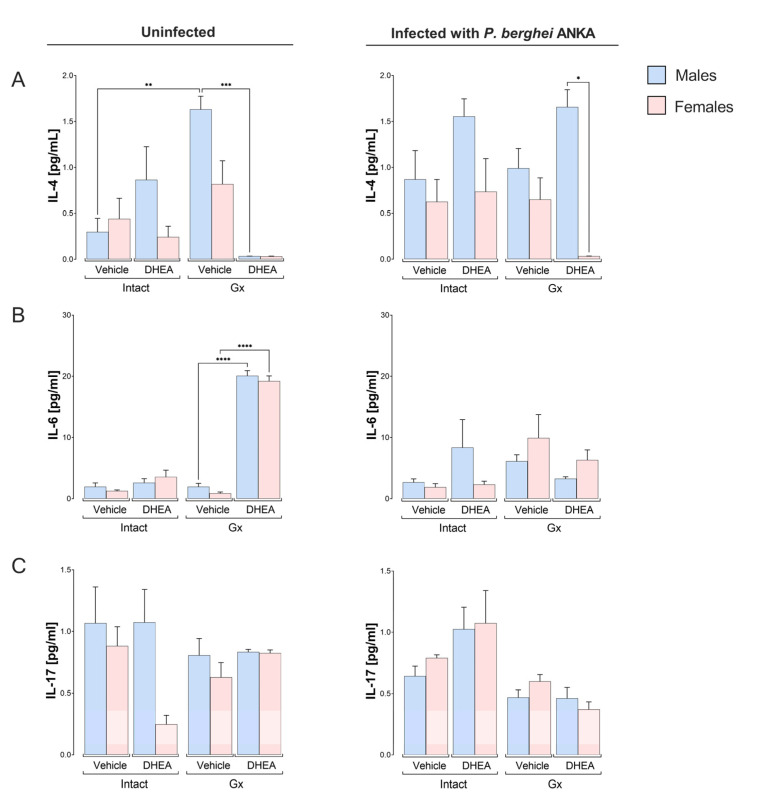
Effect of modifying DHEA levels on the plasma concentration of the cytokines IL-4, IL-6 and IL-17 in uninfected and infected male and female mice with *P. berghei* ANKA. Eight groups of male mice and eight groups of female mice were used; four groups of each sex were gonadectomized (Gx), and the other four groups remained intact. Four weeks after surgery, two Gx and two intact groups were reconstituted with DHEA for 5 days. On the day following the last administration, one group of Gx and one of the intact mice of each sex were infected with *P. berghei* ANKA, and the remaining groups were injected with PBS as infection control groups. On day 8 post infection, mice were sacrificed, and plasma was obtained to quantify the levels of the cytokines IL-4 (**A**), IL-6 (**B**) and IL-17 (**C**). Determination was performed by flow cytometry. Histograms represent the mean of each group ± SEM (*n* = 10). Lines above histograms represent significant differences between groups. Asterisks (*) indicate statistical significance between 2 groups * (*p* ≤ 0.05), ** (*p* ≤ 0.01), *** (*p* ≤ 0.001) and **** (*p* ≤ 0.0001). The significance between groups was calculated with two-way ANOVA with a Bonferroni post hoc test. The whole experiment was performed twice.

**Figure 7 ijms-24-12549-f007:**
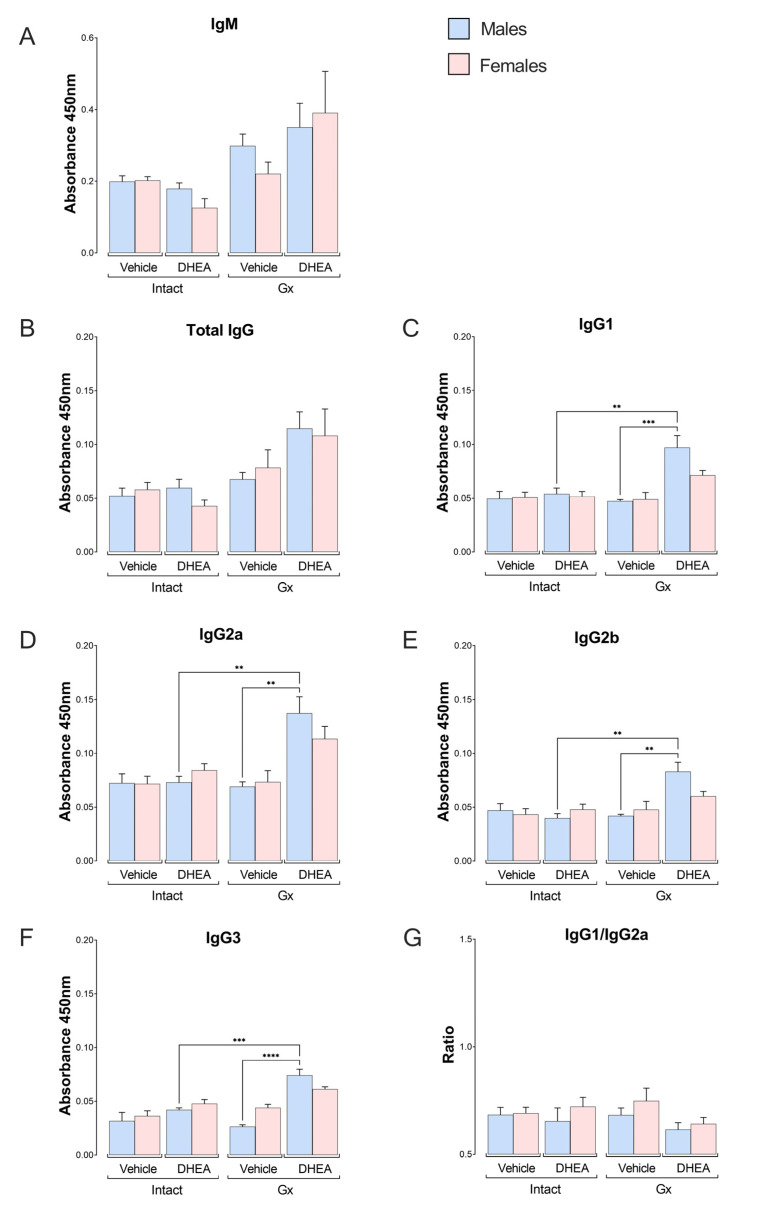
Effect of modifying DHEA levels on plasma concentration of total IgM and IgG, IgG1, IgG2a, IgG2b and IgG3 antibodies of male and female mice infected with *P. berghei* ANKA. Eight groups of male mice and eight groups of female mice were used; four groups of each sex were gonadectomized (Gx), and the other four groups remained intact. Four weeks after surgery, two Gx and 2 intact groups were treated with DHEA for 5 days. The day after the last administration, one group of Gx and one of the intact mice of each sex were infected with *P. berghei* ANKA, and the remaining groups were injected with PBS as infection control groups. On day 8 post infection, all mice were sacrificed, and plasma was separated to determine the levels of antibodies: IgM (**A**), total IgG (**B**), IgG1 (**C**), IgG2a (**D**), IgG2b (**E**), and IgG3 (**F**). Additionally, the IgG1/IgG2a ratio (**G**) was calculated. Histograms represent the mean of each group ± SEM, (*n* = 10). Lines above histograms represent significant differences between groups. Asterisks (*) indicate statistical significance between groups ** (*p* ≤ 0.01), *** (*p* ≤ 0.001) and **** (*p* ≤ 0.0001). The significance between groups was calculated with two-way ANOVA and a Bonferroni post hoc test. The whole experiment was performed twice.

**Figure 8 ijms-24-12549-f008:**
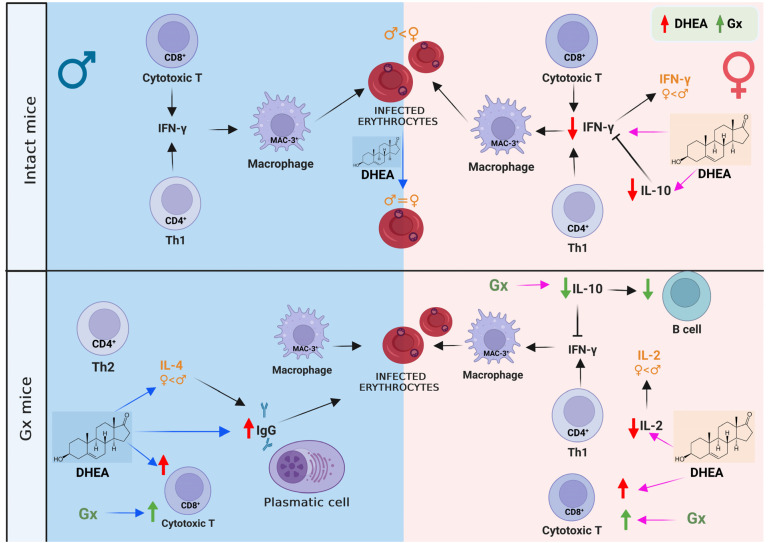
Effect of DHEA concentration on sexual dimorphism in the immune response of male and female CBA/Ca mice infected with *P. berghei* ANKA. The blue box represents DHEA concentration-induced changes in males, while the pink box represents changes in females. The green arrows indicate the effects of lowering DHEA concentration by gonadectomy and red arrows represent the effect of increasing DHEA concentration on the corresponding variables. Administering DHEA to intact males increased their parasitemia, which eliminated the dimorphic pattern present in intact mice treated with vehicle. In contrast, in the group of intact female mice, the administration of DHEA decreased the concentration of IFN-γ, which generated a dimorphic pattern and intact males presented a higher concentration of INF-γ than females under the same conditions. In addition, DHEA decreased IL-10 concentration in intact females; possibly, this is associated with IL-10 modulating IFN-γ secretion in *Plasmodium* infections. Interestingly, DHEA administration generated a dimorphic pattern in IL-2 and IL-4 concentrations in which Gx males presented a higher concentration of these cytokines than Gx females. Additionally, gonadectomy increased the number of CD8^+^ lymphocytes in both sexes; likewise, DHEA administration increased the number of this cell population. Furthermore, gonadectomy decreased the number of B cells in females, possibly by decreasing the IL-10 concentration. Reconstituting Gx females with DHEA did not restore B-cell number or IL-10 concentration. In contrast, reconstituting Gx males with DHEA increased the levels of IgG antibodies, corresponding with the higher IL-4 concentration in Gx males. The figure was created with BioRender.

## Data Availability

The raw data supporting the findings of this paper will be made available by the authors without undue reservation.

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
