# Peer review of "DHEA Induces Sex-Associated Differential Patterns in Cytokine and Antibody Levels in Mice Infected with Plasmodium berghei ANKA"

_ijms, 2023, doi:10.3390/ijms241612549_

Round 1
Reviewer 1 Report
The manuscript by Buendia-Gonzalez et al. aims to analyze the effect of DHEA on sexual dimorphism in the immune response to malaria in mice. Although the manuscript shows interesting findings regarding sex differences that should be investigated to increase the available information on the effect of sex in the immune response in infectious diseases, the authors must address some critical concerns so it can be considered for publication.
Major comments:
- Since this study was designed to compare the differences between males and females treated with or without DHEA under intact and Gx conditions, the statistical analysis for this type of study should be the 2-way ANOVA that accounts for these variables and that the authors used only for the parasitemia data but not for the rest of the figures. Can the authors explain the rationale for this? The authors should consider evaluating whether they used the correct statistical tests to analyze their data. It is essential that this part is clear enough.
- Why did the authors stop the treatment of DHEA before infection? I am curious about the rationale for this experimental design.
- Regarding their flow cytometry data, the authors refer to ‘high,’ ‘low,’ or ‘average number’ of immune cells. However, the authors never showed absolute numbers of any cell type; they showed percentages or frequencies of the cell types according to their parent gates. The authors should either show the total number or cell count if they calculated this or amend their text and clarify that they are quantifying the percentage of cells relative to some parent gate, which does not necessarily mean that there are differences in the number of total cells. This part is very important when reporting sex differences in immune cell distribution.
- Figure 1: interestingly, the DHEA levels in plasma after DHEA administration are very similar in males and females regardless of infection or gonadectomy, making me wonder whether the upper limit of detection of the EIA kit was reached. How can the authors explain this?
- Figure 2C, D: the Y-axis scale should be the same to be comparable.
- Figure 3:
i) A, B, C: the dot plot and histogram legends are tiny and difficult to read, and they should include the markers, not only the fluorochrome for each axis.
ii) they should clarify in the figure legend the parent gate they are referring to when quantifying % in all panels.
iii) The Y-axis scales should be harmonized when possible to be comparable.
iv) How can the authors explain that the CD3+ cells did not change in vehicle-treated females, but the % of CD4+ and CD8+ T cells did in the same groups?
v) The statistical analyses comparing groups of uninfected mice in Fig. 3B do not make sense (i.e., DHEA intact females vs. DHEA-Gx males and DHEA intact males with DHEA-Gx females).
- Figure 4:
i) A, B, C: the dot plot and histogram legends are tiny and difficult to read, and they should include the markers and not only the fluorochrome for each axis.
ii) they should clarify in the figure legend the parent gate they are referring to when quantifying % in all panels.
iii) The Y-axis scales should be harmonized when possible to be comparable.
- Figures 5/6:
a) Why did the authors measure cytokine levels in plasma instead of restimulating splenocytes in cell culture? Considering that the flow data they showed is from splenocytes and not from blood and that the references they cite are from in vitro cytokine production.
b) Did the authors expect to find significant differences and detectable levels of circulating cytokines in this model? What are the detection levels of the CBA kit they used? I am surprised they could find detectable levels in the plasma of cytokines such as IL-2, IL-4, or IL-17, which are known to be readily consumed by T cells after being produced in the microenvironment, especially in the plasma of uninfected mice. The authors should discuss whether the cytokine levels they measured in plasma reflect the results they would find if they had done in vitro restimulation of splenocytes.
c) Did the authors expect to see any differences in all three Types 1, 2, and 17-associated cytokines in a model of Plasmodium infection or a bias towards some specific type?
- Figure 7: Did the authors expect to find detectable levels of IgG 8 days after Plasmodium infection? The IgM and IgG subclasses levels are comparable between males and females, considering that the % of splenic B cells decreased after gonadectomy and the males did not show any changes in B cell frequency; how can the authors explain this?
Minor comments:
- Line 526: the authors should include the clone name for each antibody used.
- Lines 533-536: this sentence is duplicated from the previous lines.
- Line 543: the authors mention the presence of monocytes, but they did not show any data regarding this subset.
- Line 550: why did the authors decide to use CD16/32 to define NK cells, since the markers NK1.1/NKp46/CD49b are much better options and widely accepted in the immunology community?
- Does the Mac-3 marker have a CD molecule name? If so, the authors should include it, making it easier to identify this molecule.
- Tables 1/2: Since the authors show bar graphs for all the other data, it would be easier to visualize their statistical differences if they switched the body temperature and hemoglobin concentration tables and presented them as bar graphs (either as main or supplemental figures).
- Please state how many mice were used per group in each graph and how many times the experiments were replicated.
The manuscript must be carefully proofread for correct English grammar and vocabulary (see line 335).
Author Response
Reviewer 1
We thank you for your comments, which helped us to improve this manuscript. We addressed all your comments, and the English grammar and editing were thoroughly revised. In addition, we checked the references one by one using the EndNote program. We edited the explanation of the experimental design to make it clearer. We included explanations in the methods to make them clearer. In addition, we improved the presentation of the results and modified the conclusions.
Major comments:
- Since this study was designed to compare the differences between males and females treated with or without DHEA under intact and Gx conditions, the statistical analysis for this type of study should be the 2-way ANOVA that accounts for these variables and that the authors used only for the parasitemia data but not for the rest of the figures. Can the authors explain the rationale for this? The authors should consider evaluating whether they used the correct statistical tests to analyze their data. It is essential that this part is clear enough.
Answer. Thank you for your comment; the statistical analysis was thoroughly reviewed and repeated using 2-way ANOVA with a Bonferroni post hoc test on all figures. However, the statistical significance results did not differ from those previously described. The value of n and the number of times the experiment was performed are included in the figure legends.
- Why did the authors stop the treatment of DHEA before infection? I am curious about the rationale for this experimental design.
Answer. We interrupted the administration of DHEA one day before infection because it has been shown that with a single subcutaneous injection of DHEA, its concentration remains elevated 7 days after discontinuing its administration [1]. We administered 5 preinfection doses subcutaneously, and to corroborate that its concentration remained elevated on day 8 postinfection, we quantified its concentration.
…….
- Regarding their flow cytometry data, the authors refer to ‘high,’ ‘low,’ or ‘average number’ of immune cells. However, the authors never showed absolute numbers of any cell type; they showed percentages or frequencies of the cell types according to their parent gates. The authors should either show the total number or cell count if they calculated this or amend their text and clarify that they are quantifying the percentage of cells relative to some parent gate, which does not necessarily mean that there are differences in the number of total cells. This part is very important when reporting sex differences in immune cell distribution.
Answer. The recommendation to use absolute numbers is appropriate; however, the percentage of cells calculated from an initial dot plot is also commonly used. In our work, we performed immunophenotyping using three staining mixtures. In each one, the number of cells acquired was 10000 per sample that was selected using dot plots and was converted to percentages where the uniform region equals 100% and from there the percentage of the positive region to each fluorochrome was calculated.
The first step to analyze the results was to define the negative and positive regions for each fluorescence in each dot plot using the control of cells stained with a single antibody. Subsequently, the fluorescence of each tube containing only one fluorochrome in combination with the isotype controls was used to determine the positive region for each fluorescence.
- Figure 1: interestingly, the DHEA levels in plasma after DHEA administration are very similar in males and females regardless of infection or gonadectomy, making me wonder whether the upper limit of detection of the EIA kit was reached. How can the authors explain this?
Answer. That DHEA values are similar between males and females is to be expected given that in both sexes, DHEA is the sex steroid with the highest concentration [2].
The detection limit of the kit is 30 ng, and our results do not exceed 4 ng so the detection limit of the kit is not an issue.
- Figure 2C, D: the Y-axis scale should be the same to be comparable.
Answer. Thank you for your comment. As you suggested, in the revised version, the scale has been modified.
- Figure 3:
- i) A, B, C: the dot plot and histogram legends are tiny and difficult to read, and they should include the markers, not only the fluorochrome for each axis.
In the revised version, we improved the sharpness of the figure, the markers are written on the Y axis of the bar graph for each population, and the explanation was modified in the figure caption.
- ii) they should clarify in the figure legend the parent gate they are referring to when quantifying % in all panels.
In the revised version, we indicate within a rectangle the parent gate that corresponds to 100% of each population.
iii) The Y-axis scales should be harmonized when possible, to be comparable.
In the revised version, the Y-axis scales have been harmonized.
- iv) How can the authors explain that the CD3+ cells did not change in vehicle-treated females, but the % of CD4+ and CD8+ T cells did in the same groups?
The percentage of CD3+ cells does not change because it represents the totality of cells without considering the change in proportion originating from the CD3+CD4+ and CD3+CD8+ cell subpopulations. It is likely that the change in the proportion of CD3+CD4+ and CD3+CD8+ cells in the uninfected female mice is due to gonadectomy, which in addition to eliminating the DHEA-producing site also removes other hormones, such as testosterone and estrogens, whose functions are also immunoregulatory [3, 4]; however, this change did not modify the total number of CD3+ cells, which was part of the expected result.
- v) The statistical analyses comparing groups of uninfected mice in Fig. 3B do not make sense (i.e., DHEA intact females vs. DHEA-Gx males and DHEA intact males with DHEA-Gx females).
In the revised version, we have deleted these comparisons.
- Figure 4:
- i) A, B, C: the dot plot and histogram legends are tiny and difficult to read, and they should include the markers and not only the fluorochrome for each axis.
We improved the sharpness of the figure and added an explanation in the caption.
- ii) they should clarify in the figure legend the parent gate they are referring to when quantifying % in all panels.
In the revised version, we indicate within a rectangle the parent gate that corresponds to 100% of each population.
iii) The Y-axis scales should be harmonized when possible to be comparable.
The scales have been harmonized to make them comparable.
- Figures 5/6:
- a) Why did the authors measure cytokine levels in plasma instead of restimulating splenocytes in cell culture? Considering that the flow data they showed is from splenocytes and not from blood and that the references they cite are from in vitro cytokine production.
Thank you very much for your observation. The studies cited in the article correspond to both in vitro and in vivo studies. Tabata compared the concentrations of cytokines in patients with dermatitis and stimulated lymphocytes and described similar results [5] We considered quantifying the cytokine concentration in plasma instead of restimulating splenocytes in cell culture because plasma cytokines reflect what is occurring at the systemic level, and malaria is characterized by a systemic effect on multiple organs, resulting in severe pathologies in the brain (cerebral malaria), liver, kidney, spleen, blood, etc. Likewise, systemic inflammation mediated by proinflammatory cytokines modifies the development of Plasmodium [6]. However, for future studies, we will consider restimulating splenocytes in cell culture and compare the results.
- b) Did the authors expect to find significant differences and detectable levels of circulating cytokines in this model? What are the detection levels of the CBA kit they used? I am surprised they could find detectable levels in the plasma of cytokines such as IL-2, IL-4, or IL-17, which are known to be readily consumed by T cells after being produced in the microenvironment, especially in the plasma of uninfected mice. The authors should discuss whether the cytokine levels they measured in plasma reflect the results they would find if they had done in vitrorestimulation of splenocytes.
Answer. Certainly, we expected differences in cytokine concentrations. Using the CBA Kit (Cytometric Bead Array (CBA) Mouse Th1/Th2/Th17 Cytokine Kit (Cat No. 560485)), we previously detected significant differences in cytokine concentrations [7-9]. The detection levels in the kit used were as follows: IL-2 (0.1 pg/mL), IL-4 (0.03 pg/mL), IL-6 (1.4 pg/mL), IL-17 (0.8 pg/mL), IFN-γ (0.5 pg/mL), and TNF (0.9 pg/mL). In our recent publication, we reported that the concentration of IL-4 in the supernatant of splenocyte cultures from uninfected mice is 15 pg/mL [10], whereas in this work, we detected IL-4 concentrations in a range of 0.3 to 1.7 pg/mL depending on the group analyzed. The levels detected in the present work fall within the standard curve using the CBA kit and are approximately 10-fold lower than those detected in cell cultures. In addition, the concentrations of cytokines that we report are comparable to those reported in the plasma of malaria models [11-13]. In addition, in the immunoregulation of splenocytes, there is the possibility that several factors influence the response to antigen in in vitro cultures, such as the concentration of sex hormones, not only DHEA, which is the objective of this manuscript, and it is possible that in a cell culture, since hormones are not present at physiological concentrations, the cells do not respond in the same way to the antigen or similar to a restimulation. However, we agree that we could detect higher cytokine concentrations if we had restimulated splenocytes in vitro and for future studies we will consider restimulating splenocytes in cell culture and compare the results.
- c) Did the authors expect to see any differences in all three Types 1, 2, and 17-associated cytokines in a model of Plasmodiuminfection or a bias toward some specific type?
Answer. We expected both Th1- and Th2- type cytokines because infection with P. berghei ANKA induces a highly intense immune response in which cytokines corresponding to Th1 and Th2 responses can be detected simultaneously [14, 15]. In addition, we did not expect changes in the Th17 response because P. berghei infection did not increase the IL-17 concentration in plasma [13]. Finally, DHEA has been described to induce Th1-type responses, which by increasing the concentration of IFN-γ causes a decrease in the Th2 and Th17 responses revised in [16].
- Figure 7: Did the authors expect to find detectable levels of IgG 8 days after Plasmodium infection? The IgM and IgG subclasses levels are comparable between males and females, considering that the % of splenic B cells decreased after gonadectomy and the males did not show any changes in B cell frequency; how can the authors explain this?
This is an excellent question. Infection with P. berghei ANKA induces an extremely intense inflammatory response, resulting in the death of mice from day 9 postinfection. Due to the lethality of the infection, in our model, we cannot quantify the antibody concentration after day 8 postinfection. However, we have detected that infection increases antibody levels in our malaria model [3, 8]. We know that in general, detecting the peak of an IgG response requires more time (14 days post-infection). However, DHEA increases the concentration of IgG antibodies in mice, and it is possible to detect significant changes 7 days after the application of an influenza vaccine [17]. Related to the changes in B-cell frequencies between males and females. In both sexes, gonadectomy decreased this population; however, the difference was only significant in females. A possible explanation for the increase in Plasmodium-specific IgG1, IgG2a, IgG2b and IgG3 antibodies only in males is that by reconstituting Gx animals, particularly females, with DHEA, DHEA could be transformed by aromatase to 17β-estradiol at extragonadal sites [18], and estradiol has been shown to prevent B-cell progenitors from transforming to antibody-forming cells [19]. However, in our model, this will need to be corroborated.
Minor comments:
- Line 526: the authors should include the clone name for each antibody used.
In the revised version, the clone name for each antibody has been included in the text.
- Lines 533-536: this sentence is duplicated from the previous lines.
Thank you very much for your comment. We have eliminated the duplicated sentence.
- Line 543: the authors mention the presence of monocytes, but they did not show any data regarding this subset.
We considered macrophages to be derived from monocytes, but what we quantified was the number of macrophages, not the number of monocytes. In the revised version, we modified the text.
- Line 550: why did the authors decide to use CD16/32 to define NK cells, since the markers NK1.1/NKp46/CD49b are much better options and widely accepted in the immunology community?
Because these experiments were performed during the COVID pandemic, it was the only antibody we were able to acquire.
- Does the Mac-3 marker have a CD molecule name? If so, the authors should include it, making it easier to identify this molecule.
The Mac-3 marker is also known as CD107b, and we have included this name in the figures.
- Tables 1/2: Since the authors show bar graphs for all the other data, it would be easier to visualize their significant differences if they switched the body temperature and hemoglobin concentration tables and presented them as bar graphs (either as main or supplemental figures).
The data were represented in tables because no dimorphic effect was detected in these variables. According to your recommendations, we plotted the results and sent them as supplementary figures.
- Please state how many mice were used per group in each graph and how many times the experiments were replicated.
In each graph, we have included the number of mice per group (n=10) and the number of times the experiment was performed (2).
Comments on the Quality of English Language
The manuscript must be carefully proofread for correct English grammar and vocabulary (see line 335).
The manuscript has been carefully revised to correct errors in vocabulary and grammar.
References
- Medina, M. C.; Souza, L. C.; Caperuto, L. C.; Anhe, G. F.; Amanso, A. M.; Teixeira, V. P.; Bordin, S.; Carpinelli, A. R.; Britto, L. R.; Barbieri, R. L.; Borella, M. I.; Carvalho, C. R., Dehydroepiandrosterone increases beta-cell mass and improves the glucose-induced insulin secretion by pancreatic islets from aged rats. FEBS Lett 2006, 580, (1), 285-90.
- Elmi, A.; Galligioni, V.; Govoni, N.; Bertocchi, M.; Aniballi, C.; Bacci, M. L.; Sanchez-Morgado, J. M.; Ventrella, D., Quantification of Hair Corticosterone, DHEA and Testosterone as a Potential Tool for Welfare Assessment in Male Laboratory Mice. Animals (Basel) 2020, 10, (12).
- Aguilar-Castro, J.; Cervantes-Candelas, L. A.; Buendia-Gonzalez, F. O.; Fernandez-Rivera, O.; Nolasco-Perez, T. J.; Lopez-Padilla, M. S.; Chavira-Ramirez, D. R.; Cervantes-Sandoval, A.; Legorreta-Herrera, M., Testosterone induces sexual dimorphism during infection with Plasmodium berghei ANKA. Front Cell Infect Microbiol 2022, 12, 968325.
- Cervantes-Candelas, L. A.; Aguilar-Castro, J.; Buendia-Gonzalez, F. O.; Fernandez-Rivera, O.; Nolasco-Perez, T. J.; Lopez-Padilla, M. S.; Chavira-Ramirez, D. R.; Legorreta-Herrera, M., 17beta-Estradiol Is Involved in the Sexual Dimorphism of the Immune Response to Malaria. Front Endocrinol (Lausanne) 2021, 12, 643851.
- Tabata, N.; Tagami, H.; Terui, T., Dehydroepiandrosterone may be one of the regulators of cytokine production in atopic dermatitis. Arch Dermatol Res 1997, 289, (7), 410-4.
- Lansink, L. I. M.; Skinner, O. P.; Engel, J. A.; Lee, H. J.; Soon, M. S. F.; Williams, C. G.; SheelaNair, A.; Pernold, C. P. S.; Laohamonthonkul, P.; Akter, J.; Stoll, T.; Hill, M. M.; Talman, A. M.; Russell, A.; Lawniczak, M.; Jia, X.; Chua, B.; Anderson, D.; Creek, D. J.; Davenport, M. P.; Khoury, D. S.; Haque, A., Systemic host inflammation induces stage-specific transcriptomic modification and slower maturation in malaria parasites. mBio 2023, e0112923.
- Legorreta-Herrera, M.; Meza, R. O.; Moreno-Fierros, L., Pretreatment with Cry1Ac protoxin modulates the immune response, and increases the survival of Plasmodium-infected CBA/Ca mice. J Biomed Biotechnol 2010, 2010, 198921.
- Cervantes-Candelas, L. A.; Aguilar-Castro, J.; Buendia-Gonzalez, F. O.; Fernandez-Rivera, O.; Cervantes-Sandoval, A.; Morales-Montor, J.; Legorreta-Herrera, M., Tamoxifen Suppresses the Immune Response to Plasmodium berghei ANKA and Exacerbates Symptomatology. Pathogens 2021, 10, (6).
- Nolasco-Perez, T. J.; Cervantes-Candelas, L. A.; Buendia-Gonzalez, F. O.; Aguilar-Castro, J.; Fernandez-Rivera, O.; Salazar-Castanon, V. H.; Legorreta-Herrera, M., Immunomodulatory effects of testosterone and letrozole during Plasmodium berghei ANKA infection. Front Cell Infect Microbiol 2023, 13, 1146356.
- Salazar-Castanon, V. H.; Juarez-Avelar, I.; Legorreta-Herrera, M.; Rodriguez-Sosa, M., Macrophage migration inhibitory factor contributes to immunopathogenesis during Plasmodium yoelii 17XL infection. Front Cell Infect Microbiol 2022, 12, 968422.
- Fang, Y. Q.; Shen, C. B.; Luan, N.; Yao, H. M.; Long, C. B.; Lai, R.; Yan, X. W., In vivo antimalarial activity of synthetic hepcidin against Plasmodium berghei in mice. Chin J Nat Med 2017, 15, (3), 161-167.
- Julius, M., Cytokine levels associated with experimental malaria pathology during Plasmodium berghei ANKA infection in a mouse model. 2017.
- Helegbe, G. K.; Huy, N. T.; Yanagi, T.; Shuaibu, M. N.; Kikuchi, M.; Cherif, M. S.; Hirayama, K., Elevated IL-17 levels in semi-immune anaemic mice infected with Plasmodium berghei ANKA. Malaria Journal 2018, 17, (1), 1-12.
- Huang, B.; Huang, S.; Liu, M.; Chen, Y.; Wu, B.; Guo, H.; Su, X.-Z.; Lu, F., T cell Ig and mucin-1 and-3 in Plasmodium berghei ANKA infection. Parasitology research 2013, 112, (7), 2713.
- Kumar, V.; Rakha, A.; Saroa, R.; Bagai, U., CD4+ T cells expansion in P. berghei (NK-65) infected and immunized BALB/C Mice. J Clin Exp Pathol 2015, 5, (229), 2161-0681.1000229.
- Buendia-Gonzalez, F. O.; Legorreta-Herrera, M., The Similarities and Differences between the Effects of Testosterone and DHEA on the Innate and Adaptive Immune Response. Biomolecules 2022, 12, (12).
- Danenberg, H. D.; Ben-Yehuda, A.; Zakay-Rones, Z.; Friedman, G., Dehydroepiandrosterone (DHEA) treatment reverses the impaired immune response of old mice to influenza vaccination and protects from influenza infection. Vaccine 1995, 13, (15), 1445-8.
- Dalla Valle, L.; Couet, J.; Labrie, Y.; Simard, J.; Belvedere, P.; Simontacchi, C.; Labrie, F.; Colombo, L., Occurrence of cytochrome P450c17 mRNA and dehydroepiandrosterone biosynthesis in the rat gastrointestinal tract. Mol Cell Endocrinol 1995, 111, (1), 83-92.
- Thurmond, T. S.; Murante, F. G.; Staples, J. E.; Silverstone, A. E.; Korach, K. S.; Gasiewicz, T. A., Role of estrogen receptor alpha in hematopoietic stem cell development and B lymphocyte maturation in the male mouse. Endocrinology 2000, 141, (7), 2309-18.

Reviewer 2 Report
The manuscript "DHEA induces sex-associated differential patterns in cytokine 2 and antibody levels in mice infected with Plasmodium berghei 3 ANKA" by Buendía-González et. al, is an important study undertaking the unanswered question of sexual dimorphism and disease outcome in infectious diseases, specifically malaria infection. As the underlying mechanism of this dimorphism still needs a lot of attention, the authors attempt to study the role played by steroid hormone precursor DHEA. The study is well-designed and performed, and the results are discussed in relevant detail.
Author Response
Thank you very much for your comments.
Reviewer 3 Report
In this article, the authors investigated sexual dimorphism in immune response levels following Plasmodium berghei ANKA infection. They found that malaria parasite severity is higher in males compared to females, suggesting differences in immune response between genders. The article is well-written, and the experimental evidence supports the study's findings.
I have a few minor comments regarding the manuscript:
- In several figures, the y-axis scales are different. When comparing data, it would be helpful to use the same scale to facilitate easier comparison and identification of drastic changes. For example, in Figures 4B and 4C, the expression of different CD cells is compared, but the y-axis scales are different. Please consider modifying Figures 2C and 2D, Figures 3A, 3B, and 3C, Figures 4B and 4C, Figures 5A-5D, and Figures 6B accordingly.
- Plasmodium falciparum infects humans, whereas P. berghei is a rodent malaria parasite. There are substantial differences in the biology of these parasites in different hosts, making it challenging to directly compare immune response levels between these two studies.
- There seems to be an issue with the reference quotes in the text. Reference 42 pertains to the Plasmodium falciparum study, but the author quotes it as reference 41. Similar issues are observed with subsequent references. Please ensure the correct references are cited in the text.
- Please see the manuscript for more comments.
Overall, the article presents valuable insights into the sexual dimorphism of immune response levels following P. berghei ANKA infection. Addressing the minor comments mentioned above would further enhance the clarity and accuracy of the manuscript.

Minor editing of the English language required
Author Response
Reviewer 3
Thank you very much for your comments. A revised version of the article is attached. We considered all the comments that helped to improve our manuscript.
- In several figures, the y-axis scales are different. When comparing data, it would be helpful to use the same scale to facilitate easier comparison and identification of drastic changes. For example, in Figures 4B and 4C, the expression of different CD cells is compared, but the y-axis scales are different. Please consider modifying Figures 2C and 2D, Figures 3A, 3B, and 3C, Figures 4B and 4C, Figures 5A-5D, and Figures 6B accordingly.
Answer. Thank you for your comment. We modified the y-axis scales in Figures 3A, 3B, 4B, 4C, 5A-5D and 6B as suggested. The uninfected and infected groups display the same scale in cell populations and cytokine concentrations figures. Furthermore, the y-axis in parasitemia figures 2A-2D are homogenous.
- Plasmodium falciparum infects humans, whereas P. berghei is a rodent malaria parasite. There are substantial differences in the biology of these parasites in different hosts, making it challenging to directly compare immune response levels between these two studies.
Answer. Thank you very much for your comment. We modified the discussion, and we contrasted the effect of DHEA on the parasitemia with a similar model of murine malaria.
We revised the bibliography and fixed the numeration to include references 42, 43 and 44.
- Please see the manuscript for more comments.
Overall, the article presents valuable insights into the sexual dimorphism of immune response levels following P. berghei ANKA infection. Addressing the minor comments mentioned above would further enhance the clarity and accuracy of the manuscript.
In the revised version, the P. berghei abbreviation was used instead of Plasmodium berghei, and, we italicized the parasite specie.
